# Using Linguistic Properties of Place Specification for Network Naming to Improve Mobility Performance

**DOI:** 10.3390/s19132888

**Published:** 2019-06-29

**Authors:** Jairo López, Quang Ngoc Nguyen, Zheng Wen, Keping Yu, Takuro Sato

**Affiliations:** 1School of Fundamental Science and Engineering, Waseda University, Shinjuku-ku, Tokyo 169-0051, Japan; 2Global Information and Telecommunication Institute, Waseda University, Shinjuku, Tokyo 169-0051, Japan

**Keywords:** network naming and addressing, information-centric networking, ICN, producer–consumer mobility, mobile communications, real-time support, named-node network (3N) architecture (3NA), 3NA, NDN, ns-3

## Abstract

By considering the definitions and properties from the field of linguistics regarding place specification, a questionnaire that can be used to improve naming in networks is obtained. The questionnaire helps introduce the idea of place specification from linguistics and the concept of metric spaces into network naming schemes. The questionnaire results are used to improve the basic Information-Centric Networking (ICN) architecture’s notoriously lax network naming structure. The improvements are realized by leveraging components from the Named-Node Network Architecture, a minor ICN design, to supply the resulting network architecture with the properties the questionnaire highlights. Evaluation results from experiments demonstrate that modifying the network architecture so that the proposed questionnaire is satisfied results in achieving high mobility performance. Specifically, the proposed system can obtain mean application goodput at above 88% of the ideal result, with a delay below 0.104 s and with the network time-out Interest ratio below 0.082 for the proposed single mobile push producer, single mobile consumer scenario, even when the nodes reach the maximum tested speed of 14 m/s.

## 1. Introduction

The Information-Centric Networking (ICN) architecture is seen as a promising network model to enable environments for the Internet of Things (IoT). These environments are enabled by multiple technologies, including established research fields such as wireless sensor networks, control systems and automation. There are a series of identified requirements that need to be met for ICN to be successfully implemented for IoT [1,2,3]. One of the identified requirements is to have a naming and addressing strategy to successfully manage and communicate billions of constrained low-power devices [4]. Another identified requirement is seamless mobility as can be seen from the scenario list in [5]. Within the mobility requirements, a particularly complex requirement for ICN is the need for efficient push-based communication models [6]. A final identified requirement is the need for real-time support.

In 1982, Jerome H. Saltzer emphasized that correctly defining and naming network objects could enable seamless connectivity [7]. In the paper, the author proposed the naming of four elemental network objects (services, nodes, attachment points and paths). These names can then be leveraged through bindings that permit one to go from a service to a node, a node to an attachment point and an attachment point to a path. The important point is that a service may run at one of more nodes and may need to move from one node to another without losing its identity as a service. A similar logic is applied to nodes, with the paper explaining that a node may connect to one or more network attachment points and may need to move from one attachment point to another without losing its identity as a node. Finally, the paper mentioned that a given pair of attachment points may be connected by one or more paths, and that those paths may need to change with time without affecting the identity and the attachment points.

Saltzer’s paper was considered a fundamental work for naming network objects and was published by the Internet Engineering Task Force (IETF) as RFC 1498 in 1993 [8]. The definitions used in this work have maintained their relevance in computer network research; however, it is clear that the concept has not reached a commonly converged implementation as can be seen from the number of mobility related enhancements in [9], 18 years after the publication of Saltzer’s work. A key reason for this might be that Saltzer’s work did not offer any guideline for what properties the names of the identified objects should take.

Some researchers consider that the issues current networks are facing stem from the engineering trade-offs made when these networks were originally designed. The most famous example is the migration from IPv4 addresses using a classful network design to using Classless Inter-Domain Routing (CIDR). Although CIDR was successfully integrated into the IPv4 Internet, delaying the urgency of the problems of routing state and address depletion, it is considered a short-term solution because it does not change the fundamental Internet routing or addressing architectures [10]. The issues that necessitated CIDR have not been solved and it may be specifically because the routing and addressing architecture was not modified. As Internet usage has increased, network reliability has become more important, making fundamental changes to the network architecture unacceptable without a considerable amount of clear evidence. In such cases, a clean slate network architecture, a complete redesign of the network, its objects and components, can offer the required flexibility to modify fundamental aspects of a network architecture and test them without affecting the usage of current networks. Such attempts can provide the technology that could finally completely solve the issues or discover a method that was not apparent due to the initial unmodifiable trade-offs of current network architectures [11].

A potential clean slate network architecture is the Information-Centric Networking (ICN) architecture. ICN took an interesting approach when it came to naming and Protocol Data Unit (PDU) forwarding by utilizing names from an unmanaged, unbounded namespace for only naming information. The naming schemes in ICN, as discussed in [12], are being proposed as part of the solution to current issues facing the most promising solution for future Internet Architectures [13]. Since there are no clear guidelines as to how to use ICN’s namespace, the liberty to freely use the ICN namespace for all imaginable purposes provides an opportunity to re-evaluate our understanding of the role of networking naming on network performance. Work under prominent ICN architectures has already demonstrated that naming is a factor in enabling low latency connectivity [14,15].

### 1.1. Our Contributions

This article explores whether there can be a convergence in network naming strategies to improve performance, particularly under mobility. The key contributions of this article are the following:The summary of the properties for place specification from linguistics that will help simplify future discussions about network naming schemes. Of particular importance is the fact that place specification is a relator that requires the definition of a figure *F* and a ground *G* (Section 2).The creation of a questionnaire to help network architects think about what needs to be named to be found in a network and the properties that can be integrated into these names (Section 4).The highlighting of the improvable points in the ICN architecture’s main naming strategy by an evaluation utilizing the questionnaire (Section 5).The modification of the ICN architecture to satisfy the questions not satisfied in the initial evaluation (Section 6).The value of the questionnaire is demonstrated with the modified architecture obtaining a goodput above 88% of the ideal result, a delay below 0.104 s and a network time-out Interest ratio below 0.082, regardless of node speed on the mobility framework for the single mobile push producer, single mobile consumer scenario, clearly outperforming the original architecture (Section 8).

The rest of the article is organized as follows: The current understanding of the codification of place specification in languages used to simplify future discussions is realized in Section 2. Section 3 explains the basic components required to understand how names are utilized in an ICN for PDU forwarding. This section includes some of the mobility issues identified by researchers when using the ICN approach, as well as the MAP-Me mobility enhancement available for ICN architectures. Section 4 utilizes the linguistic understanding presented in Section 2 to create a questionnaire to evaluate network naming schemes and their forwarding strategies. The evaluation of the ICN procedures explained in Section 3 are done in Section 5. In Section 6, modifications to the naming and forwarding schemes are proposed to improve the network architecture. Section 7 presents the experimental framework and the parameters used when evaluating the modifications to ICN. Section 8 explains the evaluation of the proposals’ performance. The results are discussed in Section 9. Finally, in Section 10, the conclusions are presented.

## 2. Properties in Languages for Space Specification

Network names and their properties have a direct impact on how easily a labeled object can be located within a network. This assertion is supported by the multi-disciplinary research done by Steven C. Levinson and his collaborators [16]. Their work defines the primitives and the methods used in human language for place specification. In their work, they describe a fundamental set of properties that can be found in all researched human languages that facilitate the transmission of location information. This article will focus on three properties which are the definition of location as a spatial relator, the ubiquitous topological information available in human languages and the necessary inclusion of the temporal dimension for motion description.

### 2.1. Spatial Relators

The first important uniformity found in languages is that place specification is always expressed as a spatial relator *R*, a binary relation between a *figure* (*F*) and a *ground* (*G*). *F* is the object to be located and *G* is the search domain on which *F* can be located. Place specification always entails the creation of a binary relation between an *F* and a *G*. Most languages implicitly use the surface of the Earth as the ground *G* of the largest scope, even though such a definition is generally unnecessary in daily communication ([16] Chapter 2.3).

### 2.2. Topology

Topology studies geometrical properties that persist in a constant state under transformation or ‘deformation’. In language, topological prepositions help describe spatial relations of proximity, order, enclosure and continuity of objects in a given space. Examples for these spatial relations in English include the use of semantic notions like ‘near’, ‘at’, ‘between’ or ‘in’. Topological prepositions are always available and are vital in place specification as they help narrow down the search domain *G* to communicate place specification ([16] Chapter 3.3.3).

### 2.3. Temporal Dimension

Motion events can be located as wholes, meaning that languages can generally utilize all the same resources as when describing a static location. Motion is more complex than static location because it necessarily involves the temporal dimension. Integrating time automatically permits the inclusion of information about the change of location, how the location changed, the medium being used to create that change and the instrument enabling the motion. Linguists have not been able to find a methodology that can encompass all the possible codifications of temporal dimension information when locating a particular motion event ([16] Chapter 3.5).

## 3. ICN Architecture

ICNs were created in a desire to move from host-centric networking to information-centric networking. These networks function by labeling information. This naming of information means that every piece of information that can be requested has a label that will be referred to in this paper as an ICN name. Communication is driven by consumers who request information utilizing an Interest PDU with the desired ICN name. Nodes who participate in this network, each identified as a Content Router (CR), forward the Interest PDU until the corresponding information to the ICN name is found. The information is returned in a Data PDU which uses the ICN name. To carry out Interest and Data PDU forwarding, each CR maintains three data structures, a Pending Interest Table (PIT), a Forwarding Information Base (FIB) and a Content Store (CS). The PIT stores all the Interests that a CR has forwarded but not yet satisfied. Each PIT entry keeps track of the ICN name solicited along with its incoming interface. The FIB keeps entries which map ICN names to interfaces. These entries are leveraged by algorithms, named forwarding strategies, to determine where to forward Interest PDUs by their ICN names.

The upstream and downstream flow of Interest and Data PDUs are shown in Figure 1 and Figure 2. When an Interest PDU arrives at a CR, the CR checks the CS for information matching the ICN name used in the Interest PDU. If a match exists, the CR returns the information in a Data PDU on the interface from which the Interest PDU arrived. Otherwise, the ICN name is looked up in the PIT and, if a matching entry exists, the incoming interface identifier is aggregated to the PIT entry. A forwarding strategy is then utilized to forward the Interest PDU. When a Data PDU arrives, the CR finds the matching PIT entry and forwards the Data PDU to all downstream interfaces listed in the PIT entry. Once the Data PDU is forwarded, the PIT entry is removed and the information in the Data PDU is stored in the CS for future use. The maintenance and cooperation of CSs in an ICN are some of the key elements to obtain a high user Quality of Experience (QoE). How CSs cooperate is a topic that continues to present interesting solutions [17,18].

Another key element to obtain a high user QoE in ICN is the selection of an appropriate forwarding strategy. Forwarding strategies determine where received Interest PDUs should be transmitted. Usually, the forwarding strategy is linked with the FIB, but this varies depending on the ICN implementation. Since forwarding strategies are usually based on the mapping of ICN names to possible transmission locations, Section 3.1 and Section 3.3.1 are dedicated to describing these strategies in more detail.

For Named-Data Networking (NDN), one of the main ICN implementations, there is no mention of a network naming structure for the architecture’s main namespace [19]. A structure is implied with the use of URLs in its presentation. There is also an understanding that individual links to connect CRs would be managed by existing networks. In subsequent work, the main namespace design only assumes hierarchically structured names. Namespace management is declared not to be part of the architecture, leaving it up to users to define namespace structures and management for their needs. According to [20], naming is considered to be the most important part of NDN application design, as it enables the architecture’s advantages. However, a naming guideline is not presented. An analysis of ICN naming, utilizing an interpretation of Saltzer’s work, can be found in [12]. Once again, a naming guideline is not presented. With the ICN architecture reimplementing network naming strategies from previous network designs, such as in [21], it is unclear if known naming issues are not being introduced into the network architecture.

The lack of a guideline for mapping the ICN namespace to existing networks causes issues for broadcast mediums. In these mediums, the fallback transmission method is to broadcast the information. This is not energy efficient [22,23], which causes drawbacks for wireless network use cases. At the same time, the lack of rules for the assignment of names in ICN offers us the chance to modify and test network naming on the architecture.

### 3.1. Smart Flooding (SF)

Smart Flooding (SF) is a forwarding strategy for ICN architectures described in detail in [24]. In SF, interfaces are given one of three states, green if it can bring back information, yellow if it is unknown whether the interface may bring back information and red if it is clear that the interface cannot bring back information. When an ingress Interest PDU gets passed to the FIB, the PDU is forwarded through the highest ranked green interface available. After a predetermined amount of time, if the PIT entry for the related ICN name continues to be unsatisfied, lower ranked green interfaces are tested and the initially highest ranked interface is demoted. If the PIT entry is unsatisfied, the process continues, in descending order, through all green, yellow and red interfaces available. When a PIT entry exhausts the FIB entries available for the ICN name, then a special Interest PDU, called a Non-Acknowledgement (NACK) PDU, is transmitted in a downstream manner. Neighboring nodes whose ingress interface receive this NACK PDU are automatically set to red state for the related ICN name. Should the NACK PDU set all FIB entries for the ICN name to the red state, then the NACK PDU is again forwarded downstream. On the other hand, when the ICN named information is found and sent downstream in a Data PDU, the downstream nodes automatically promote the related ingress interface to the green state.

### 3.2. MAP-Me enhancements (MM)

An unintended consequence of the ICN architecture is that, when the producer is mobile, Interest PDUs may end up being forwarded toward sectors of the network where the producer is no longer attached. As only consumers can retransmit Interest PDUs, depending on how these are handled, significant delay is expected [25].

To overcome this disadvantage, the authors in [26] create a protocol that handles producer mobility events by dynamically updating the FIB. When a producer changes point of attachment, it sends a modified Interest PDU, named MAP-Me-IU that acts as an update protocol. The MAP-Me-IU includes the ICN name prefixes that the producer is offering, along with a sequence number. This method ensures that the newest MAP-Me-IU is utilized to update the FIB. Since the MAP-Me-IU leverages an Interest PDU, when the PDU is received by a CR, it can be forwarded upstream in the same way as a normal Interest PDU. This permits the MAP-Me-IU to be forwarded via all interfaces from which Data PDUs related to the ICN name have been received. After forwarding the MAP-Me-IU, the interface on which the MAP-Me-IU was received is used to update the CR’s FIB. The ingress interface for the ICN name is enabled if the sequence number on the MAP-Me-IU is newer than previously seen MAP-Me-IU at the CR. A significant drawback of this protocol is that it relies on the existence of a routing protocol responsible for creating and updating the FIBs of CR. When MAP-Me-IU PDUs are not being transmitted, the network relies on standard ICN methods to update the FIB.

### 3.3. 3N Architecture

The Named-Node Network Architecture (3NA) is an attempt to improve the delay and goodput of ICN for producers and consumers. 3NA does not have specialized nodes to support mobility, as all nodes have the same structures and capabilities. 3NA capable nodes differ from a common ICN node in three key ways. First, all nodes handle two additional independent namespaces. Second, there are two new structures to utilize the two new namespaces. Finally, 3NA creates two sets of new PDUs, mechanism PDUs and data transmission PDUs to leverage the new namespaces [27,28].

The first independent namespace adds unique labels to nodes belonging to the network. This namespace is called the Node name (3N) namespace. Names from this namespace are called 3N names. The 3N namespace attempts to be a metrizable topological space and consists of a single, complex, multi-level structure into which all 3N names fit. The 3N names follow strict naming instructions and their assignation in the network is highly regulated to maintain the namespace as a metrizable topological space.

The second independent namespace adds labels to a node’s Point of Attachments (PoAs) to the network. This namespace is called the PoA namespace and consists of the names typically used to identify interfaces on a physical transmission medium. Names belonging to this namespace are called PoA names. PoA names must only be unique among the end points of a shared communication medium. The namespace mappings for 3NA can be seen in Figure 3.

The Node Name Signature Table (NNST) is an additional structure to the common ICN architecture. This structure keeps a mapping between 3N names and PoA names. The mapping is used to create direct communication on a physical medium between nodes.

A second addition to the common ICN architecture is the Node Name Pair Table (NNPT). This structure maintains a 3N name to 3N name mapping. The mapping is used to check whether nodes with a 3N name have been updated due to a change of the nodes’ point of attachment to the network. In most cases, this type of reattachment makes the node use a different 3N name to maintain the properties of the names in the 3N namespace.

Updates to the NNST and NNPT structures are done by signaling. Signaling is made possible by a set of PDUs called mechanism PDUs. Mechanism PDUs are utilized to enroll, dis-enroll, re-enroll nodes into the network and to update this information to the network. The mechanism PDUs for 3NA are summarized in Table 1. The NNST is signaled whenever a node enrolls into the network via the use of the EN, OEN and AEN PDUs. Whenever a node dis-enrolls from the network, the NNST is signaled via the use of the DEN and ADEN PDUs. All re-enrollments are signaled to the NNST via the use of REN, OEN and AEN PDUs. The NNPT is signaled to update 3N name mappings whenever a node re-enrolls into the network via the use of the INF PDU. A simplified enrollment procedure is summarized in Figure A1. A simplified re-enrollment procedure is summarized in Figure A2. A more detailed explanation of the procedures are explained in [27,28].

The common ICN PIT was also modified to aggregate the 3N names of the nodes sending Interest PDUs. This modification allows Data PDUs to be sent directly to nodes by using 3N names. If 3N names are not used in 3NA, the network performs exactly like the ICN architecture as described in Section 3.

#### 3.3.1. 3NA Forwarding Strategies

3NA utilizes a set of PDUs called data transmission PDUs to make use of 3N name information in the architecture’s forwarding strategies. These PDUs include 3N name information in its fields and permits the encapsulation of ICN PDUs. The encapsulation allows the inter-operation of 3NA networks with other ICN implementations. The data transmission PDUs defined in 3NA are summarized in Table 2. More details about the data transmission PDUs are available in [27,28].

3NA relies on SF to forward Interest PDUs. Compared to a pure ICN, the 3NA gives all nodes participating in the network a name, permitting the aggregation of those 3N names within the PIT. When a Data PDU is sent downstream, a 3NA node will trigger a search for the aggregated 3N names in the PIT entry. Before the 3N name is used in the delivery of a Data PDU, a check in the NNPT is performed to make sure that the 3N name being used is the newest available 3N name. The NNST is then used to figure out which path to use for the forwarding process by comparing the destination 3N name with the PoA to 3N mapping maintained in the structure. This search results in a PoA name and an associated interface that would best fit the requirements. The use of 3N names in the Data PDU means that, in 3NA, one is routing directly on the 3N names and can leverage the mathematical properties of the namespace. The resulting forwarding strategy is called Smart Flooding with 3N names (SF 3N). If 3N names are not used when using SF 3N, the forwarding strategy performs exactly like SF (Section 3.1).

One very important use case that 3NA can support is forwarding Data PDUs without the need of PIT entries. When nodes using 3NA enroll into the network, they are given a 3N name. Due to the properties of 3N names, if the offered name is (5.0.0), one knows that the node who offered it is (5.0). With this information, a 3NA node is capable of creating a DO or a DU PDU to encapsulate Data PDUs to reach specific named nodes. Normally, the CS in ICN is updated only when a related PIT entry is available. The usage of DO or DU PDUs offers the possibility of a different trigger flow which can be used for producers to push the desired Data PDUs to a given node. This easily enables the support for push producers.

## 4. Use of Linguistic Properties in Network Naming Schemes

Section 2 highlighted three properties of languages that are available in all researched languages. These properties are vital in being able to correctly convey place specification information. The availability of these properties implies that even simpler human made languages, such as the set of questions and answers used in computer network place specification, could also have these traits.

Since humans can communicate location information among members of the species in a highly sophisticated manner, one can leverage the work and findings of Levinson and his collaborators to evaluate forwarding strategies in ICN. The evaluation holds merit because what forwarding strategies attempt to do is to guide the PDUs, in other words, how to best reach the location for a network name after calculating where it is located. If one assumes that the place specification strategies used in human languages can be adapted for forwarding strategies, it would be extremely important for properties found in human languages to be available in the namespaces utilized in computer networks.

This reasoning motivated us to elaborate a simple questionnaire to determine if the properties of human languages appear in the names used in a computer network. To emphasize the complexity of place specification in language and how it links to the questionnaire, the following scenario is proposed. Imagine you are with two friends, Taro and Hanako, with which you share an apartment. Somewhere parked outside are also three bicycles that each have their own lock with a key. Although everyone has their own bicycle, it is common for the bicycles to be shared. Now imagine that you want to go out and are looking for your particular key. A simple question directed at Taro and Hanako could be, “Have you seen my bicycle key?”. Already at this point in the conversation, there has been a definition of what one is wanting to find; a key, and a specific key; your bicycle key. This process is essential to begin place specification (Question 1).

Since you are in a relatively organized apartment, an answer from Hanako could be, “I think it is next to the other keys”. Hanako’s answer is nuanced as it is implying that all keys, including your bicycle key, are in the same location. Notice that Hanako does not single out the types of keys. In a normal apartment, there is the definite possibility of multiple keys. Hanako also does not actually offer a location for your specific key. She is talking about the proximity of your bicycle key with the other keys. This is possible because a topological space can be constructed among the keys in the space in which we live (Question 3).

You instinctively move over to the place where keys are normally stored. Although Hanako has not defined a location explicitly, it is there implicitly. In some apartments, the place where the keys are stored is at the apartment entrance, or a bowl on a particular table, or on some particular key rack within the apartment. Regardless of the actual location, both Hanako and you are instinctively aware of the search domain, its size and how far away it is from where the conversation is taking place. Both Hanako and you have defined a ground (Question 4).

Aware that Hanako is right that all keys are usually at this pre-designated storage, you are amazed to find that your bicycle key is not there. All the other identified keys you know about are there. At that moment, Taro, who has been quiet throughout the seconds this conversation has taken, speaks up and says to you, “Your key was there this morning. I took it by mistake. It is in my bag.” The search domain has been re-defined and it is different from what Hanako and you had defined. This is possible because both the search domain Hanako and you defined, and the search domain Taro and you defined can actually be mapped to an implicit search domain of bigger scope, one of which can be the apartment (Question 4). More impressively, such simple language has introduced the temporal dimension to the conversation specifying that the key was at some location and has indeed been moved (Question 2). After this exchange, hopefully you can get your bicycle key from Taro’s bag, or, more likely, Taro will get it for you, as a bag can be a restricted search domain.

In the scenario above, the characters rely on topological descriptors to tell you what might be next to your bicycle key or what it is in. Section 2.2 described how Levinson and his collaborators found that topological properties permeate all codifications of place specification in languages. When dealing with computers networks, one has to limit the scope of the questions to a metric space because there is no easy way to have a computer understand the *notion* of ‘near’ or ‘far’. Metric spaces have the advantage that they satisfy the conditions for topological spaces, allowing us topological reasoning, while giving us numerical values that a computer can handle to deterministically define conditions related to ‘near’ and ‘far’. A metric space is formally defined as an ordered pair (M,d) where *M* is a set and *d* is a metric on *M*. In other words, a function
d:MxM→R
such that for any x,y,z∈M, the following holds:d(x,y)≥0,d(x,y)=0⇔x=y,d(x,y)=d(y,x),d(x,z)≤d(x,y)+d(y,z).

Function *d* is commonly referred to as a distance function.

With the definitions established, the questionnaire can be described below:


**Question 1.**


What is the name *n* of the object you want to find?


**Question 2.**


Is *n*’s location updated over time?


**Question 3.**


Is *n* part of a metric space $M_{N},N\in N$?


**Question 4.**


For every possible object $n\in M_{N}$, can these objects be located in a single defined metric space $M_{N-x},x\in N$?

The questionnaire is of no use unless one can concretely define what it is that one is attempting to find. This is the objective of Question 1, as it forces one to define the figure *F*. Without defining figure *F*, one cannot contemplate the definition of a shared ground *G*. In ICN, a consumer only requests ICN names, so the consumer depends on the network to find where the related chunks are located to satisfy the request. The assumption is made in this dicussion that what needs to be located are chunks related to an ICN name which are the size of the network’s Maximum Transmission Unit (MTU).

Question 2 is derived from the knowledge of place specification in languages when motion is involved, as described in Section 2.3. Although there are no definite methods to codify motion in languages, the locations $g_{n}$ of a name *n* must be somehow maintained and updated over time.

Question 3 asks if a name *n* being used in a network naming scheme can be placed in a particular set of names and if these can be categorized so that they can be organized and ordered by a distance function. The ability to do these actions permits forming a notion of ‘near’ between two names belonging to the same set.

Question 4 deals with the scope of the ground *G* used by a name *n*. Section 2.1 discussed the implicit use of the Earth’s surface as the ground *G* with the largest scope in human languages. From the same section, it is also known that, in human languages, place specification relies on establishing a binary relation between a defined figure *F* and a defined ground *G*, even if *G* is only implied by context. This means that the name *n* referred to in Question 3 is the *F* in the binary relation. This also implies that, in network naming structures, a *G* always exists. An interesting implication is that, if it is known that the relation between a *F* and a *G* is binary, a recursive place specification can be performed such that the $g_{1}\in G_{1}$ of a $F_{1}$ can also be seen as the $F_{2}$ of a $G_{2}$, where $G_{2}$ has a different scope when compared to $G_{1}$. This implies that $F_{1}$ could also be located in $G_{2}$. What Question 4 aims to determine is whether all *n* used can be located in some *G*, regardless of the number of necessary steps to obtain that mapping. If *G* is a metric space and *F* is specified as $g_{1}\in G$, then one can designate a search area, defined by a distance, in *G* for other $g^{\prime }$. If these $g^{\prime }$ exist, due to place specification being a binary relator, it is known that a valid $g^{\prime }$ may lead to a $F^{\prime }$. This is an extremely useful way to search for related objects without needing to know the names for $F^{\prime }$.

## 5. Evaluation of ICN Forwarding Strategies Utilizing the Developed Questionnaire

Utilizing the questionnaire developed in Section 4, the questions are evaluated for the ICN forwarding strategy presented in Section 3.1. From the main ICN explanation in Section 3, it is known that there is no naming structure for ICN names and that there is no defined distance between two particular names. Hence, for all forwarding strategies that use ICN names without modifications, Question 3 is unsatisfied. This applies for Smart Flooding (SF) and Smart Flooding with MAP-Me enhancements (SF MM).

Section 3 also states that, in the ICN architecture, the mapping of ICN names is done to interfaces. From the discussion in Section 2.1, this would mean that the figure *F* would be the ICN names and their ground *G* would be the interfaces. In ICN, these interfaces are normally labeled. Even if the naming of interfaces used numbers, which would satisfy the metric space definition, the scope of the naming would be local to a particular node. This means that, for all forwarding strategies that use the basic ICN naming, mapping ICN names to interfaces via the PIT and FIB, without modifications, Question 4 is unsatisfied. This applies to SF and SF MM.

SF has the NACK PDU to avoid attempts to forward PDUs via interfaces through which there is historical information that suggests that a particular ICN name will not be found. Since NACK PDUs update the FIB over time, there is an update of the location of ICN names over time in SF. Although SF does not pay special attention to chunks, due to all non-aggregatable ICN names being tracked in the FIB, Question 2 is satisfied.

SF MM attempts to speed up the update of ICN name to interface mapping by ensuring certain interfaces have a green state for future Interest PDUs. In the update, only the prefix of all the ICN names provided by the producer are updated, pointing all matching ICN names to one given interface, possibly undoing the work done by the SF’s NACK PDUs. We have established in Section 4 that what needs to be tracked over time are the chunks related to an ICN name. Due to this conflicting update, Question 2 is only partially satisfied.

Section 3.3.1 briefly described how 3NA links Smart Flooding to 3N names via the use of specialized PDUs when using its SF 3N forwarding strategy. 3NA does not directly modify the ICN namespace, meaning that Question 3 continues to not be satisfied. Although the assigning of names in 3NA is done maintaining particular namespace properties, the ICN namespace does not map to 3N names. 3NA could permit the creation of metric spaces for all ICN namespaces as the 3N namespace covers the whole network. This makes the 3N namespace a good candidate as the definition of a ground for place specification. Considering these points, Question 4 is only partially satisfied. With some minor modifications, it is possible for this question to be satisfied. This possibility will be leveraged in Section 6.2 when discussing modifications to the network naming schemes. SF 3N leverages SF, satisfying Question 2.

## 6. Improvements Derived from the Developed Questionnaire

The previous section has shown that the original ICN architecture does not satisfy Question 3 or Question 4. It was also shown that, for SF MM, Question 2 is only partially satisfied due to different objects being updated over time. Therefore, modifications that permit an ICN architecture to satisfy Question 2 and Question 3 are proposed in Section 6.1. The modifications used to satisfy Question 2 and Question 3 are integrated into 3NA to satisfy Question 4. The details of the integration with 3NA are detailed in Section 6.2. Table 3 summarizes the questionnaire results for all the forwarding strategies utilized in this article.

### 6.1. Smart Flooding with MAP-Me and Part Routing (SF MM PR)

Frequently used names in ICN are suggested to be formatted in a hierarchical manner to be able to aggregate the names when used in the FIB and PIT structures. Leveraging this information, one can further strengthen the naming rules to fit a hierarchy with *⌀* as the root vertex of a tree graph and each edge being traversed adding a ‘/’. For this article, each vertex in the resulting ICN name graph has a string of any type. An example of the resulting graph is shown in Figure 4. From graph theory, it is known that one can define a graph distance d(u,v) where u,v are vertices of the graph and the distance is defined as the minimum number of edges needed to be traversed to get from vertex *u* to *v*. This definition permits the definition of a metric space.

Although ICN names can be easily mapped to a graph using this strategy, within an Information-Centric Network, a particular ICN name would be divided into chunks to transmit the content in pieces throughout the network as mentioned in Section 4. Leveraging this characteristic of ICNs, a numbered label is added to identify the chunks. These identified chunks of information related to an ICN name are called Parts. We assign numbers to the Parts to create another metric space utilizing the one-dimensional Euclidean distance to define the distance between Parts. The definitions of ‘≥’ and ‘<’ for natural numbers can then be used to establish intervals and aggregate the Parts present at any node.

This new information can be easily leveraged by the MAP-Me enhancements. In the MAP-Me-IU, the producer would add the next ICN name Part $p_{j}$ for information transmission. When a ICN CR receives this information on a given interface, it only changes the state of the interface to green for Parts $\geq p_{j}$. The modification of the MAP-Me update system to utilize Part information is called Part Routing. When Smart-Flooding, MAP-Me enhancements and Part Routing are used, the resulting forwarding strategy is summarized as SF MM PR. This forwarding strategy would satisfy Question 2 as the MAP-Me enhancements would update the location of Parts directly. Question 3 is also satisfied as the ICN namespace is a metric space.

### 6.2. Smart Flooding with MAP-Me, Part Routing and 3N Names (SF MM PR 3N)

A lot of modifications to the ICN architecture would be required for every ICN name to be located in a metric space. In contrast, the 3NA, as defined in Section 3.3, avoids mapping ICN names to 3N names. Without any modification, the 3NA would not satisfy Question 4. The 3NA specification states that, when nodes are enrolled in a network, they obtain a 3N name from a namespace that maintains a metrizable topological namespace. From the definition of 3N names, one can construct a graph as shown in Figure 5. 3N names can be mapped to the graph in which vertices are labeled with a number and each edge traversed to get to a vertex adds a ‘.’ to the final name. Similar to Section 6.1, one can define the graph distance as the metric for 3N names, permitting the namespace that holds all 3N names to be considered a metric space.

To create a mapping from ICN names to 3N names, the changes made to ICN names in Section 6.1 are leveraged. This permits the aggregation of ICN names by both hierarchies and the intervals for Parts, and have this aggregated information be mapped to 3N names. The modification for Parts Routing made in Section 6.1 was used in the MAP-Me-IU PDU to update FIB interfaces only for particular Parts. A similar modification to the MAP-Me-IU can be made to include 3N name information.

There is one key difference in the use of 3N names in MAP-Me-IU. To satisfy Question 4, the 3N namespace is the metric space to which one can map every ICN name. For mobile nodes that are pushing content, the mapping of ICN names to the mobile node 3N name, which would constantly change, does not seem as efficient as mapping the ICN names to the 3N name of the node to which the content was initially pushed. For this article, the discussion of ICN name to 3N name mapping is limited to this configuration for use in the modified MAP-Me-UI. This mapping of a Parts interval to a 3N name satisfies the definition of place specification as seen in Section 2.1. With this modification, Question 4 can be satisfied.

There is an additional advantage of using 3N names in downstream transmission of Data PDUs. In a generic ICN, the Data PDU is transmitted through the interface which initially saw an Interest PDU. However, due to movement, mobile nodes constantly change the point of attachment to the network. In 3NA, each node has a 3N name which can be aggregated in the PIT structure. The 3NA also keeps a dynamic mapping via its NNPT structure to point to the latest 3N name linked to a node that initially enrolled using a particular 3N name. If the node correctly enrolls, dis-enrolls and re-enrolls to the network, then every PDU that uses any of the 3N names obtained by the node in the first or subsequent enrollments will reach its destination.

When Smart-Flooding, MAP-Me enhancements and Part Routing are used with ICN names mapped to 3N names, and the resulting forwarding strategy is denoted as SF MM PR 3N. In this forwarding strategy, aggregated ICN names are mapped to 3N names along with all the other 3N name update mechanisms from the 3NA. This forwarding strategy permits us to satisfy the whole questionnaire.

## 7. Mobility Framework

To assess the impact of using a correct naming structure can have on network mobility, nnnSIM [27,28,29] an implementation of 3NA, on top of the ns-3 network simulator framework [30] is leveraged. The mobility framework was taken from [28] with a few parameters updated. The scenario consists of a single Graph of a Network (GN). The scenario includes seven wireless access points (WAPs) in a regular hexagon pattern, connected in a hierarchical fashion as is shown in Figure 6. The WAPs are 215 m apart and utilize the default WiFi implementation in the ns-3 framework for version ns-3.29, where each WAP has its own SSID. ICN CRs are connected via wired connections.

The scenario tests deterministic mobility which relates to fixed public transport systems, such as trains, trams and monorails, with producers and consumers riding inside these vehicles. In the scenario, there is only one consumer and one producer, mimicking public transport systems where users connect to in-vehicle access points. The vehicles themselves have only one network interface to attach to the network and do not have CSs. The scenario is simple, making it easily reproducible and verifiable. The simplification also permits the mobility framework to quickly determine if the advantages seen in the results for the single mobile consumer with a mobile push producer use case were directly caused by the modifications made to the network naming strategies. The main disadvantage of such a simple scenario is that it does not test for multiple vehicles accessing content.

Note that 3NA capabilities derived from using 3N names are only enabled when the consumer node uses these capabilities. The network nodes obtain 3N names from the central WAP. When the network utilizes 3NA functions, all 3NA related buffering of PDUs, particularly during node name changes, network enrollment, dis-enrollment and re-enrollment, are turned off to be able to compare forwarding strategies. When a consumer does not use 3NA functionalities, the wired network nodes act as regular ICN CRs using SF as described in Section 3. Another important parameter is that the size of the CS is 10 million Parts or 10.24 GB. At the Interest PDU generation rate shown in Table 4, the producer would only generate 0.3125 GB of data during the simulation, ensuring that the network could always count on CRs to respond for data quickly if the information was available.

We choose a simplified handover procedure for mobile nodes. When a node travels 100 m, it checks the RSSI signal strength of the neighboring WAPs. The nodes automatically migrate to the WAP with the best signal at that point in time. When using 3NA, migration means executing signaling procedures before and after changing WAP. The whole handover process performs appropriately during the simulation. For these procedures, the network updates its data structures, particularly the NNPTs and the NNSTs [27].

Mobile nodes move in the simulation at constant velocity with speeds from 1.4 m/s to 14 m/s, increasing by intervals of 1.4 m/s. These speeds represent inter-city velocities from average human walking speed to slightly higher than the last mile average car driving speeds. These speeds were updated due to newer average speeds published in [31]. The nodes trace the perimeter of a regular 230 m sided hexagon as is shown in Figure 6. The mobile nodes never share the same SSID as they are always at opposite sides of the hexagon. The cyclical pattern of movement permits us to compare the impact of speed and constant SSID changes on the forwarding strategies.

In 2016, the second largest data streaming rate for video on the Internet was 2500 kbit/s [32]. This value was updated from the original framework to reflect the increase of video bit rate since the framework was first published. A parameter that was made explicit was the number of Interest retransmission attempts. It became important to explicitly mention the upper limit, especially when considering the retrieval of more time sensitive information. An upper limit of 4 was chosen, considering the Interest retransmission timeout. Another parameter that was made explicit was the Data PDU payload size. As was briefly mentioned in Section 4 regarding the definition of a Part, the Part was considered to be the size of the MTU. Considering the size of the ICN headers and the 3NA headers, the definition of a payload size of 1024 Bytes was proposed and tested. The final parameter modified was the application start time for the producer and the consumer. It was necessary for the push producer to not push information exactly at the same time that the consumer requested the same information. This discrepancy was needed to see how ICN forwarding strategies dealt with Parts being dispersed in the network and not necessarily on the path to the producer. A low number of 20 s after the push producer began transmitting content was chosen.

The producer node is an 3NA enabled edge pushing content node. This means that the producer has a 3N name and continuously pushes its content to the WAP to which it is connected. The producer does not respond to any requests for content from the network. The use of a 3N name enables us to push content without having to modify ICN’s Interest/Data PDU protocol flow as established in [20].

All of the network parameters used in the simulation are summarized in Table 4.

## 8. Performance Analysis

The mobility framework is used to test SF (Section 3.1), SF MM (Section 3.2), SF MM PR (Section 6.1) and SF MM PR 3N (Section 6.2). The analysis focuses on how using different network naming schemes affects the performance metrics described below:**Mean delay experienced by the consumer**: A low overall delay would ensure a pleasant viewing experience for streamed content.**Mean goodput experienced by the consumer**: For the consumer to have a good viewing experience, the goodput performance should be maintained close to the actual data streaming rate of the content. We chose to compare goodput percentage to the desired ideal because it summarizes the application-level throughput. Goodput is calculated by using the formula shown in Equation (Equation 1):
(1)$$\begin{array}{c} \#\;\mathrm{of}\;\mathrm{PDUs}\;\mathrm{received}\;\mathrm{by}\;\mathrm{the}\;\mathrm{Rx}\\ \mathrm{Goodput}=\frac{\mathrm{in}\;\mathrm{sequence}\times \mathrm{PDU}\;\mathrm{payload}\;\mathrm{size}}{\mathrm{Time}}. \end{array}$$**Network timed out Interests to satisfied Consumer Interests ratio**: Interest timeout in ICN means that the network can track a consumer’s request by using the PIT to remove Interest requests that cannot be satisfied after a specified timeout period. If a forwarding strategy has low network Interest time out to satisfy consumer Interests ratio, it means that a generated Interest is generally locating the requested Data. This also implies that the consumer does not put a considerable burden on the whole network by forwarding the Interest via unnecessary interfaces.

### 8.1. Evaluation Results

The mean network delay for the four forwarding strategies are shown in Figure 7. For these results, there is a clear difference in making a network naming scheme satisfy Question 2. SF MM (Question 2 =△) follows the same trend as SF (Question 2 =◯). SF manages better results, making SF MM the overall worst performing forwarding strategy. Although the difference between SF and SF MM (Question 3 =×) and SF MM PR and SF MM PR 3N (Question 3 =◯) is not large before the nodes start moving above 7 m/s, it becomes more than half afterwards. The maximum tested speed of 14 m/s shows a remarkable difference in delay of more than 400% between these sets of forwarding strategies. SF MM PR 3N (Question 4 =◯) obtains the overall lowest mean network delay. The delay for all forwarding strategies is always lower than 0.4 s.

The mean goodput compared to the ideal goodput of 2500 kbit/s for the four forwarding strategies are shown in Figure 8. Once again, a clear difference in making a network naming scheme satisfy Question 2 is observed, with SF MM obtaining the overall worst performance. A clear difference in making a network scheme satisfy Question 3 is also noticed. SF MM PR 3N, which satisfies Question 4, is always 4% better than the next forwarding strategy, SF MM PR. For SF MM PR 3N and SF MM PR, the goodput is maintained above 85% of the ideal, regardless of the moving speed of the nodes. For SF and SF MM, the goodput starts above 87% at 1.4 m/s, then fluctuates and quickly decreases to below 80% of the ideal goodput when the nodes start moving faster than 7.0 m/s.

The ratio of network timed out Interests vs. satisfied consumer Interests for the four forwarding strategies are shown in Figure 9. There is, once again, a large difference in making a network naming scheme satisfy Question 2. SF MM obtains the highest overall ratio. SF, which satisfies this question, obtains better results. A large difference is also seen in making a network naming scheme satisfy Question 3. SF MM PR 3N has the lowest ratio of timed out Interests vs. satisfied consumer Interests which is relatively stable and close to zero, regardless of the node movement speed. A similar pattern is observed for SF MM PR. SF and SF MM both start with a low ratio but rise linearly as the speed of the nodes increase.

From the evaluation results, it is clear that SF MM PR 3N, which satisfies Question 2, Question 3 and Question 4, clearly outperforms all the other forwarding strategies. SF MM PR 3N obtains the lowest delay, best goodput and lowest Interest timeout to satisfied consumer Interest ratio regardless of node movement speed. SF MM PR, which does not satisfy Question 4, has a lower performance in goodput and slightly higher average delay than SF MM PR 3N. This forwarding strategy outperforms SF and SF MM, with a clear difference seen in goodput. SF, which does not satisfy Question 3 or Question 4, has a marked lower goodput performance, higher delay and higher Interest timeout to satisfied consumer Interest ratio than SF MM PR and SF MM PR 3N. SF MM, which doesn’t satisfy any question, obtains the worst overall results.

## 9. Discussion

In a generic ICN, the flooding mechanism, forwarding an Interest PDU through all available interfaces except for the ingress interface, is the only defined place specification mechanism. This method is unable to determine what related Parts are necessary or where they are located. In other words, ICN defines a figure *F*, but it gives no clue as to how to find Parts that are ‘close’ to that initial *F*. This means that SF only updates the location of the one particular ICN name in the Interest PDU that caused the initial flooding. A related downstream Data PDU would enable the interface for future use, but it assumes that subsequent related Data PDUs are stored in one of the nodes that are connected by the newly enabled interface. Specifically, the naming scheme does not have any topological properties. This feature of the Data PDU is also shared by the MAP-Me-IU. The location of future ICN name chunks are assumed to be locatable using the interface that leads to the mobile producer. This is an assumption that can be easily wrong when a producer is moving fast. This assumption gets worse when the producer is pushing content while moving. In the mobility framework used, the consumer begins to ask for an ICN name 20 s after the producer has begun to push content, meaning that a currently enabled interface in the FIB might not correspond to the actual Part that the consumer has requested. The results of SF MM are consistent with this interpretation, as SF MM (Question 2 =△) obtains overall worse results when compared with SF (Question 2 =◯). As the speed of the producer node increases, related Parts are divided into smaller intervals on the edge CR CSs, with bigger gaps between two complete intervals on any given CS. The results worsen when the node reaches a speed that not only makes the related Part intervals smaller, but has the intervals be on nodes that are further apart in the GN. This in turn causes more flooding because the network has no way of locating the Parts in advance.

As the results in the previous section have shown, being able to ask *where* a particular piece of an ICN name, which was named a Part (Question 3 =◯), is located, automatically offers us results that greatly improve delay, goodput and the Interest timeout to satisfied consumer Interest ratio. This improvement is maintained even if the ground *G* is of limited scope (Question 4 =×). The limited scope of the ground *G* is not as problematic in the mobility framework due to the fact that the wired network creates a connected tree where one interface has only one possible destination. Due to the properties of a tree graph, like the one used in the GN, there is negligible difference between keeping the 3N name of an edge node updated for a Part interval and keeping the interface that leads to the edge node updated. If the requirements of one’s network can be satisfied with the delay and goodput levels shown in the results, then enabling a larger scope for ground *G* seems unnecessary. This is a point worth considering, particularly when considering the number of modifications required to satisfy Question 4.

If one wishes to further improve delay and goodput, the previous results show a significant improvement when the naming scheme defines a ground *G* of larger scope that is also a metric space (Question 4 =◯). It is important to remember that in the mobility framework, all buffering was disabled and PDUs had to be forwarded in a reactive manner. This means that the improvements observed come only from PDUs being forwarded using updated 3N name (ground *G*) information. Equally important is that downstream Data PDUs generally reach the consumer because the 3NA NNPT keeps the list of the latest 3N name being used by nodes. This particular feature accounts for the difference seen between SF MM PR and SF MM PR 3N.

The introduction of the chosen properties in language for place specification in network naming only scratches the surface of the types of optimizations that can be realized when integrating this knowledge to improving aggregation and forwarding schemes. The results of simply defining such properties in an ICN with no fixed naming scheme and leveraging them in the simplest manner offer significant improvements for the single mobile consumer scenario. The improvements observed warrant further research with multiple consumers and a variety of GN for future network deployment.

By utilizing the ideas of place specification in language, one can avoid common pitfalls when talking about network identifiers and locators. By first specifying what figure *F* is, what the ground *G* is on which one can locate *F*, what scopes these namespaces have and how they are generated, and more concise discussions about network naming schemes can occur.

## 10. Conclusions

By understanding that location is a binary spatial relator for which topological information is always available in human languages and that the temporal dimension must be included when dealing with motion, a simple questionnaire has been created that can verify if these properties are present in a network naming scheme. The questionnaire forces the user to determine what object in the network needs to be found, if the object and its location is tracked over time, whether that object’s name is part of a metric space and whether the object can be located in a metric space. After answering the questionnaire for an ICN architecture, the knowledge of properties in language for place specification has been utilized to modify the architecture to satisfy the questionnaire. The modified ICN, which leverages from 3NA designs, demonstrates a significant improvement of network delay, goodput and Interest timeout to satisfied consumer Interest ratio for the single mobile push producer, single mobile consumer scenario, even when the nodes reach the maximum tested speed of 14 m/s. The improvements obtained for the proposed mobility framework using the modified ICN implementing place specification properties warrant further research of these properties on more complex network graphs with multiple producers and multiple consumers before being considered for real network deployment. 

## Figures and Tables

**Figure 1 sensors-19-02888-f001:**
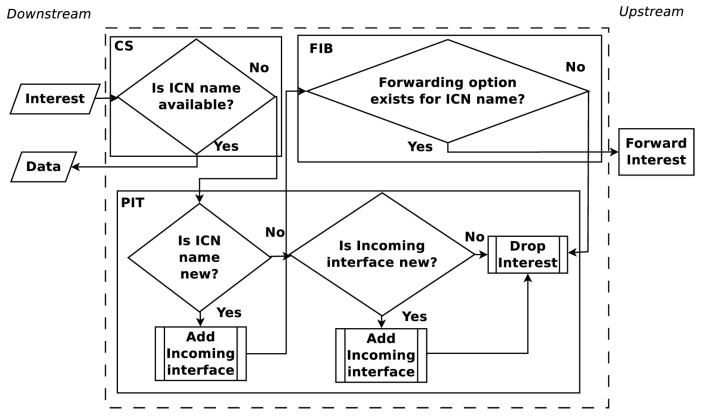
Forwarding process for Interest PDUs at an ICN node.

**Figure 2 sensors-19-02888-f002:**
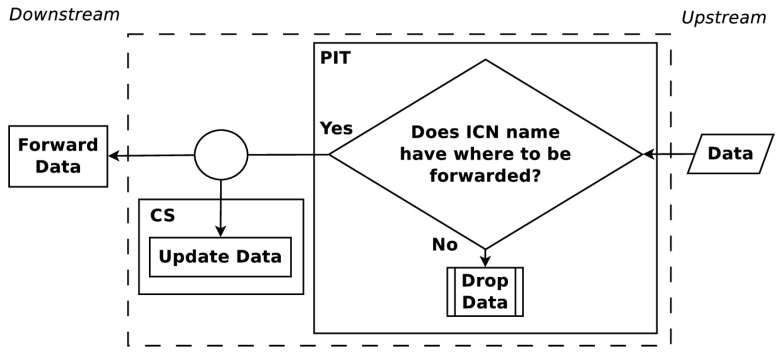
Forwarding process for Data PDUs at an ICN node.

**Figure 3 sensors-19-02888-f003:**
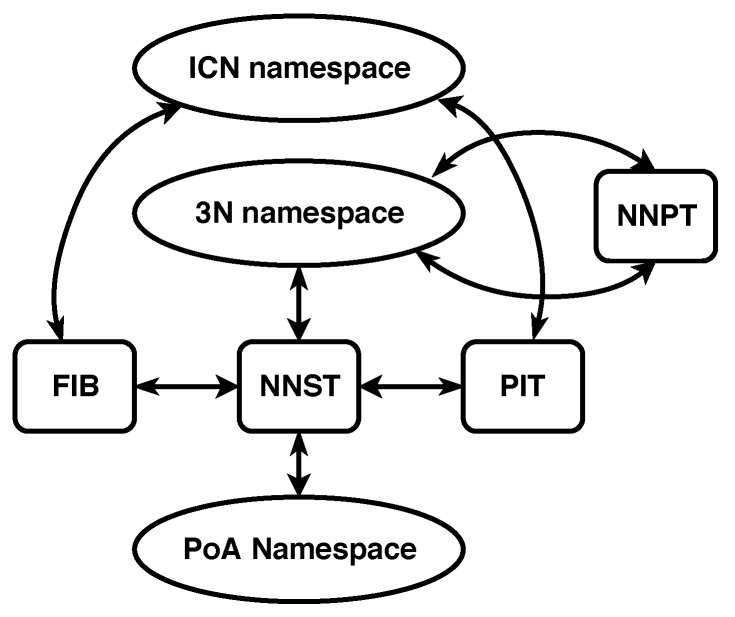
3NA namespaces and mappings.

**Figure 4 sensors-19-02888-f004:**
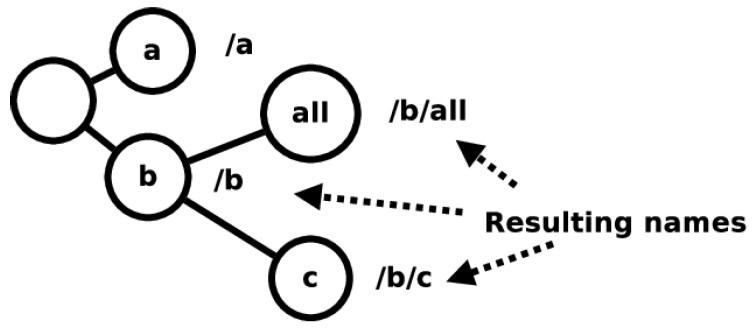
A graph for ICN names.

**Figure 5 sensors-19-02888-f005:**
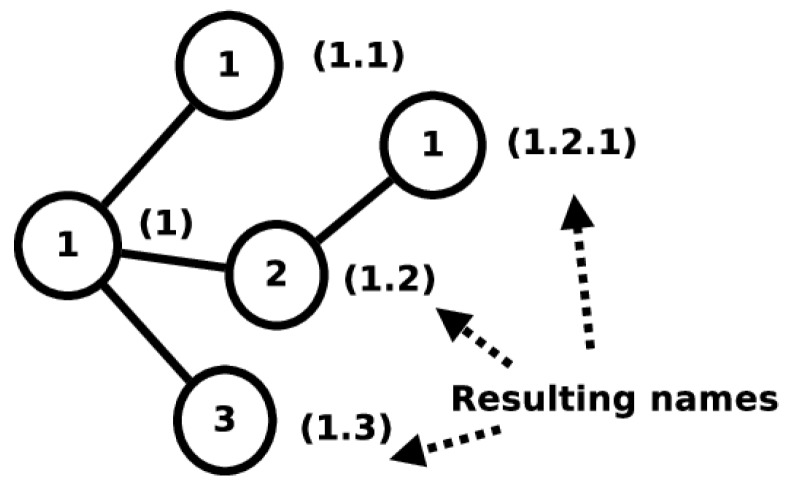
A graph for 3N names.

**Figure 6 sensors-19-02888-f006:**
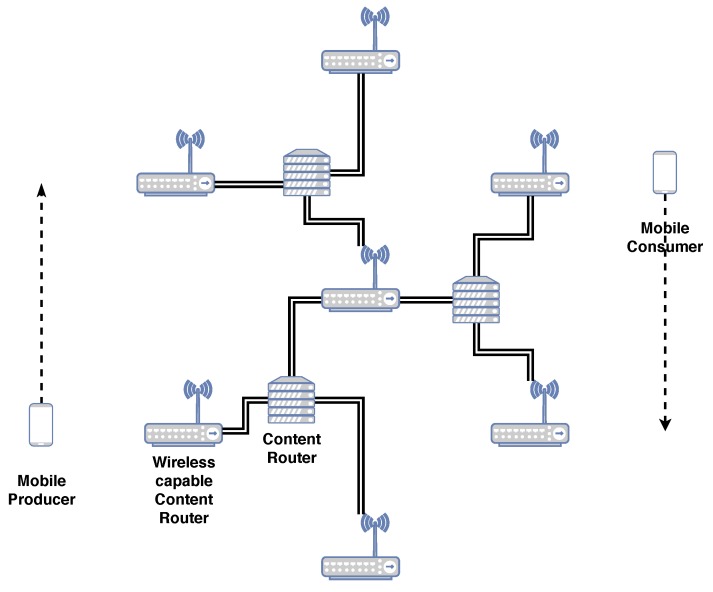
Graph of the network used for the mobility framework.

**Figure 7 sensors-19-02888-f007:**
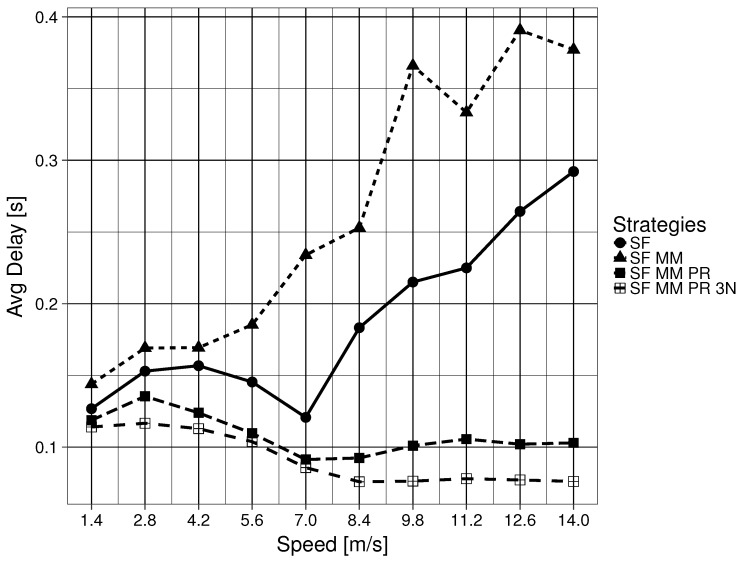
Mean network delay for consumer for forwarding strategies.

**Figure 8 sensors-19-02888-f008:**
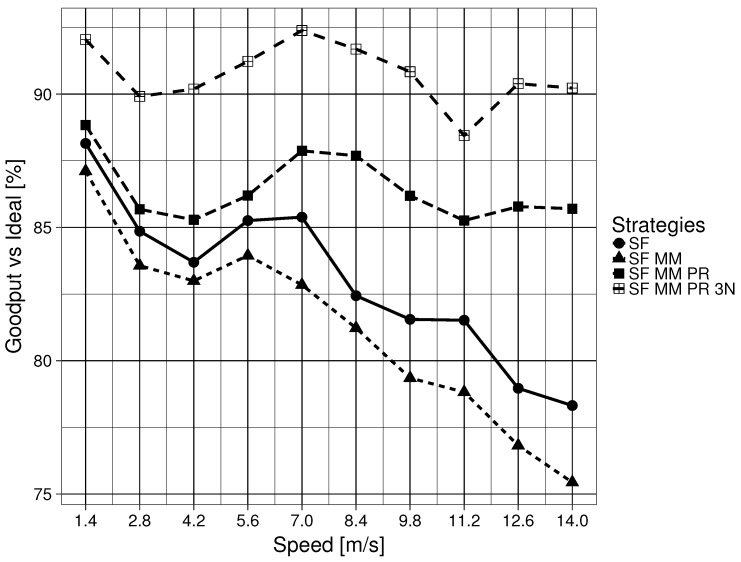
Mean goodput percentage of ideal for forwarding strategies.

**Figure 9 sensors-19-02888-f009:**
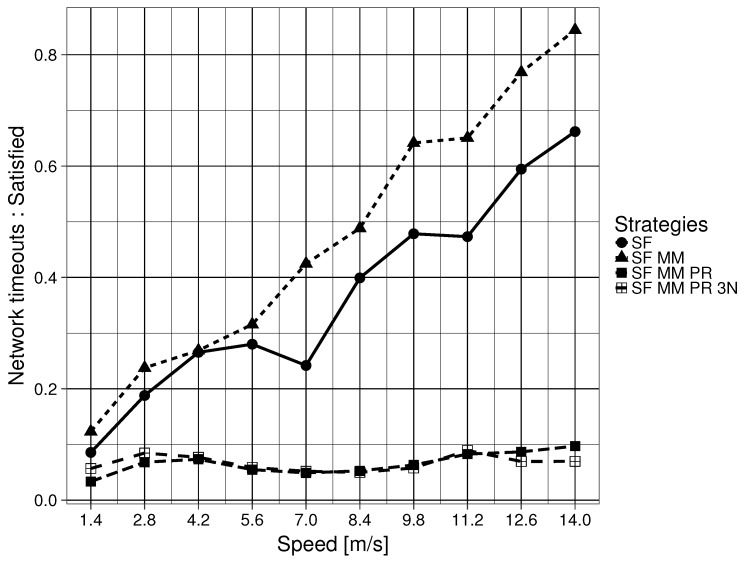
Timed out Interests to satisfy consumer Interests ratio for forwarding strategies.

**Table 1 sensors-19-02888-t001:** 3N architecture mechanism PDUs.

PDU Name	Description
EN	Enrolls nodes into a sector
OEN	Offers a name to an enrolling node
AEN	Acknowledges the enrollment of a node into a sector
REN	Re-enrolls a node into a new sector while the node still has a valid 3N name
DEN	Dis-enrolls a node a from a sector
ADEN	Acknowledges the dis-enrollment of a node from a sector
INF	Informs sectors about nodes obtaining new 3N names

**Table 2 sensors-19-02888-t002:** 3N architecture data transmission PDUs.

PDU Name	Description
SO	Includes only Source node’s 3N name
DO	Includes only Destination node’s 3N name
DU	Includes the Source node’s along with the Destination node’s 3N name

**Table 3 sensors-19-02888-t003:** Questionnaire evaluation results for ICN forwarding strategies.

Strategy	Question 2	Question 3	Question 4
SF	◯	×	×
SF MM	△	×	×
SF 3N	◯	×	△
SF MM PR	◯	◯	×
SF MM PR 3N	◯	◯	◯

◯ = Satisfied, △ = Partially satisfied, × = Unsatisfied.

**Table 4 sensors-19-02888-t004:** Parameters for the graph of the network.

Parameter	Value
Data PDU payload size	1024Bytes
Interest PDU generation rate	306 PDUs/s
Interest retransmission timeout	50 ms
Number of Interest retransmission attempts	4
PIT Entry Timeout	1 s
Content Store size	10 million Parts
Consumer start time	20 s after producer
3N name lease time	300 s
Video bit rate	2500kbit/s
Link delay (wired)	5 ms
Bandwidth/Link capacity (wired)	100 Mbps
Link properties (wireless)	Constant Speed Propagation
	Three Log Distance Propagation
	Nakagami Propagation
Simulation time	1000 s
Mobile node speed	1.4, 2.8, 5.6, 7, 8.4, 11.2, 12.6, 14 m/s

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
