# Peer review of "Using Linguistic Properties of Place Specification for Network Naming to Improve Mobility Performance"

_sensors, 2019, doi:10.3390/s19132888_

Reviewer 1 Report

General comment: Although the pioneering work of John Day described in the PNA book is referred to in the acknowledgements, the authors don’t explain how their approach compare to RINA (Recursive InterNetwork Architecture), the evolution of the concepts described by Mr. Day in PNA. 

I would recommend the authors to include a section in which they compare their 3NA approach to RINA, stating major differences and similarities.

If this issue is resolved, as well as the detailed comments below, I would recommend the paper for publication.

Detailed comments:

Line 23 -> What needs to communicate are the applications contained in those devices; the devices are just containers for the apps. This is an important distinction for naming and addressing, since it is key to recognise what network objects need to be named, the paper should make it clear and avoid the pitfall of stating that “low-power devices” need to communicate with each other.

Line 25 -> Do all IoT applications need to communicate with all other IoT applications in the world? Or just with a small subset? In other words, what is the scope of communications? For many use cases the requirement will be of private communications between a relatively slow number of application instances. Maybe this should be stated in the introduction also.

Line 32 -> it would be important to state the fact that Saltzer didn’t recognise that there can be multiple paths between a pair of nodes, and hence routes should be computed as a sequence of nodes - and not as a sequence of points of attachment (otherwise, the number of routes grows combinatorially with the number of paths and hops in the network).

Line 62 -> How does “information” relate to the four concepts Saltzer describes as key for naming in networks (services, nodes, attachment points and paths)? From the analysis at the beginning of section 3 it seems to me that information is the data that services (or applications, another word for services) work with. ICN is only naming “information” (i.e. application data) and interfaces. According to Saltzer, it is missing service and node names. This situation doesn’t look very promising for a network architecture: ICN lacks 2 of the 4 key concepts, and uses one (“information”) which is not relevant for networking according to Saltzer. I think that these points should be made clear in the paper, since it uses Saltzer in the introduction but doesn’t analyse ICN naming in the context of Saltzer’s model.

This analysis should also help the reader understand why ICN will have problems with routing in general, specially if the use case involves “complications” such as mobility.

Line 116 -> And Saltzer made it clear that what matters is “service-centric” (or “application-centric”) networking. 

Line 137 -> Talking about QoE, if all a user can do is request an “Interest”, how does the user express what is the quality of service required for his/her communication? I know it is not the matter of this paper, but does ICN support multiple QoS classes? Could different forwarding strategies be potentially used per QoS class?

Line 203 -> If a name is topological (such as the 3N Name) it is also usually called an “address”; the authors may want to point this out.

Line 195 -> This is a comment for section 3.3. I think that the authors should go back to Saltzer and state that they are trying to improve ICN by adding one of the two missing concepts (nodes, services): nodes. So, instead of having “information” directly mapped to “interfaces”, no we will have “information” mapped to “nodes” and “nodes”  mapped to interfaces. Still not completely following Saltzer’s model (we would need services instead of information), but much closer.

Line 234 -> This comment is for section 3.3.1. If I got it right, 3NA conceptually is basically about adding a layer of “nodes” between the “information” and the “interfaces”. Nodes will be the one doing the data forwarding based on “topological node names” (i.e. node addresses) and *not information*. Of course this will enhance the performance and scalability of the routing system. BUT, can we call this ICN anymore? If the whole ICN idea is about “forwarding data in the network based on information labels” and now we are “forwarding data in the network based on node names”, where is ICN?

Line 240 -> Why always use flooding for maintaining the distributed directory (“information” to node name bindings)? In many scenarios this won’t be a scalable approach (agree that it will be in some others). So why not considering other alternatives (hierarchical directories, DHTs, etc.)

Line 350 -> This comment is for section 5. The questionnaire seems to be a useful tool, but just by looking at ICN in the light of Saltzer’s model, it is clear that routing/forwarding without using service names and node names will have problems. 

Line 417 -> Even if the ICN namespace is a metric space, the “distance” between two given ICN names says nothing relative to their attachment in the network. In other words, the ICN namespace may be relevant to help the application itself locate information (e.g. as to how is the application information internally organised), but it doesn’t help the network to understand where the information is stored.

Line 617 -> I think that an important conclusion is missing -> ICN routing does not work well for mobility. The “modified ICN” that the authors propose is not ICN anymore, since routing is not done on information names but on node names (which is the key differentiation of ICN). Again, that should not be a surprise, since according to Saltzer ICN is missing service names and node names. Doing “network routing” based on “information names” is not a good idea, and the authors are providing significant evidence of it. I think that this should be made mode clear, since it is an important result that other colleagues in the networking field should be aware of.

Author Response

The authors thank the reviewer for their time in commenting about our manuscript. Please see the attached PDF addressing all the reviewer's comments point by point.

Reviewer 2 Report

The discussed problem is interesting and timely in the context of ICN. However, the paper fails to properly present the proposed solution. The paper is tiring and almost half of it is introductory information. But even when the proposed solution is presented it is not done in a clear and comprehensive way. Maybe some real world examples would have been helpful.

Furthermore, the evaluation of the proposed solution is superficial and unrealistic:the authors assume a single consumer, a single producer and CRs capable of storing the whole workload. Scenarios with multiple endpoints and realistic workloads are expected from a journal paper. 

Author Response

The authors thank the reviewer for their time in commenting about our manuscript. Please see the attached PDF addressing all the reviewer's comments point by point.

Round  2

Reviewer 2 Report

I am not convinced by the authors response wrt to the evaluation. Neither convinces me their statement in the paper "The use of one consumer and one producer without CS was chosen to limit the noise that multiple consumers and producers with differing CS policies could create" If in a scenario where multiple consumers exist, the advantages of the proposed solution are not noticeable, then may be the solution is not useful in a realistic set up. 

Author Response

The authors thank the reviewer for their time in commenting about our manuscript. Please see the attached PDF addressing all the reviewer's comments point by point.

This manuscript is a resubmission of an earlier submission. The following is a list of the peer review reports and author responses from that submission.

Round  1

Reviewer 1 Report

I cannot really understand why a paper on mobility would be relevant for a journal named Sensors, as there are no sensors involved. Assuming that the editors do not care about sensors, should the paper be accepted? The answer is a resounding no, as starting from the abstract the paper is full of overblown generalities of no practical significance. The authors draw upon networking history and linguistics to plug their architecture, called 3NA, as a supplement to NDN, although a) they do not really present 3NA, b) 3NA does not seem to be that much of an ICN architecture (what with node and interface names), c) they lack a basic understanding of ICN ( NDN <> ICN), d) they do not even understand part of NDN (NDN does not use the term interface), e) they propose an extension to NDN implemented as an extension to 3NA and f) they compare it against itself. Now, if 3NA was an architecture widely known in the ICN community, they could have dropped the entire NDN part and made a reasonable paper. But it is not, therefore it has to be explained if it is to be used in a paper and improved upon. On the other hand, if the basis for finding NDN insufficient is that it does not fulfill some arbitrary criteria set by the authors, then I find it hard to be persuaded that we should replace it with another architecture that also does not meet them.

Author Response

The authors thanks the reviewer for the comments. The authors have responded to the comments in the attached PDF.

Reviewer 2 Report

The research topic of this paper is good.

The descriptions of the evaluation of the proposed method, however,  lacks of detailedness.

For the following parameters used in the simulations, the authors must describe the adequacies the selected values.

1) CS size

2) Interest packet generation rate

3) Video bit rate

For the above 1), the reviewer wonders that why the so huge size are used.

In the real situation, any consumer can not occupy such large buffer.

Under the parameters used in the evaluations of this paper,  buffer overflow will never occur due to the CS size.

The reviewer could not agree with such condition.

In addition, this paper evaluated the performance only the case that single consumer/single producer scenario.

For the mobile and wireless situations, it is not realistic.

By the way, the study discussed in this paper has a high relationship with a study discussed in the following paper.

[P1] Seamless Mobility in ICN for Mobile Consumers with Mobile Producers

Jairo LÓPEZ  Takuro SATO  Trans. IEICE EB,  pp.1827- 1836, 2017.

However, the evaluation parameters used in both papers are different.

The authors must explain why the those parameters are changed.

[Minor comment]

Several figures used in the paper are most of the same with the figures appeared in the literature [P1].

Hence, the authors should clear the copyright issues among two publishers (journals).

Author Response

(The authors gave the same response as above.)

Reviewer 3 Report

Authors propose a novel way of enhancing mobility performance in ICN networks using a linguistic approach (metric spaces) to evaluate and enhance different ICN approaches. 

Please address the following aspects within the paper:

- The title can be improved, it should emphasize more the novelty of the study and the use of linguistics for solving a networking problem. 

- include reference to Levinson in line 83.

- I could not see quite clearly the intention of section 3.1 - please make it clearer in the first paragraph of that section. I was expecting an analysis of related work on naming as well. Or at least reasons why you are not including it.

- maybe you can include an illustrative example at the beginning of section 4 to make clearer the role of the questionnaire. 

- Questions from the questionnaire should be in the form of questions. As they are now they seem more premises than questions. 

- consider rephrasing the questions for the questionnaire so that they reflect the networking jargon which can make easier the general understanding of the assessment framework you're proposing. For example, for question 1 you seem to be asking implicitly "Can a name be mapped into a location?", of course, you propose broader reachability for the "question" but it can be improved by introducing the purpose with a simplified question.   

- please check the spelling in general. But specifically, address line 267 “the the”, 269 “in in”,  463 “fir”, 472 “that that”, 810 “we can to locate”. There might be a few others I've missed. 

- please check statement of line 271. 

- also, relate the paragraph 808-811 to the questionnaire.

Author Response

The authors thank the reviewer for the comments. The authors have responded to the comments in the attached PDF.

Round  2

Reviewer 3 Report

You have improved the paper with respect to the previous version, however you need to pay attention to spelling, typos and grammar throughout the whole document. Please carefully check the text within the following line numbers:

10 (word repetition, "environment"), 25 (and ... and), 42-43 (seen ... seen), 63 (and can been ?!), 186 (no definition of CR), paragraph 239 (SF is smart flooding or smart forwarding?), 357 (first sentence is incomprehensible), 361 and 366 (what does "to interface mapping" mean?), 368 (is a positive? please amend), 378 (it is not clear what "answering with delta" means...), 419 (check grammar, it should be "part of the reason why"), 

Other aspects to improve:

* Figures A3-A5 do not contribute much to the paper, I'd suggest to take them out. 

* Add a legend to Table 3.

* It is not completely clear why the use of SF MM PR would respond to question 4, as described in  line 408.

* In line 429 you mention that 3N names have relatively longer lifespan and you have never mentioned it before. Please make sure this statement is coherent with what it is previously said in other sections.

* line 438 lacks clarity: "each node a 3N name which ..." ?